# Modelling the Reallocation of Time Spent Sitting into Physical Activity: Isotemporal Substitution vs. Compositional Isotemporal Substitution

**DOI:** 10.3390/ijerph18126210

**Published:** 2021-06-08

**Authors:** Gregory J. H. Biddle, Joseph Henson, Stuart J. H. Biddle, Melanie J. Davies, Kamlesh Khunti, Alex V. Rowlands, Stephen Sutton, Thomas Yates, Charlotte L. Edwardson

**Affiliations:** 1Diabetes Research Centre, University of Leicester, Leicester LE5 4PW, UK; g.j.biddle@lboro.ac.uk (G.J.H.B.); jjh18@le.ac.uk (J.H.); melanie.davies@uhl-tr.nhs.uk (M.J.D.); kk22@le.ac.uk (K.K.); alex.rowlands@leicester.ac.uk (A.V.R.); ce95@le.ac.uk (C.L.E.); 2NIHR Leicester Biomedical Research Centre, Leicester LE5 4PW, UK; 3Health Sciences, University of Leicester, Leicester LE1 7RH, UK; 4The Centre for Lifestyle Medicine and Behaviour, School of Sport, Exercise and Health Sciences, Loughborough University, Loughborough LE11 3TU, UK; 5Centre for Health Research, University of Southern Queensland, Education City, Springfield Central, QLD 4350, Australia; stuart.biddle@usq.edu.au; 6Leicester Diabetes Centre, University Hospitals of Leicester, Leicester LE5 4PW, UK; 7NIHR Collaborations for Leadership in Applied Health Research and Care (CLAHRC) East Midlands, Leicester LE5 4PW, UK; 8Behavioural Science Group, Institute of Public Health, University of Cambridge, Cambridge CB2 0SR, UK; srs34@medschl.cam.ac.uk

**Keywords:** sedentary behaviour, physical activity, time use, cardiometabolic health

## Abstract

Isotemporal substitution modelling (ISM) and compositional isotemporal modelling (CISM) are statistical approaches used in epidemiology to model the associations of replacing time in one physical behaviour with time in another. This study’s aim was to use both ISM and CISM to examine and compare associations of reallocating 60 min of sitting into standing or stepping with markers of cardiometabolic health. Cross-sectional data collected during three randomised control trials (RCTs) were utilised. All participants (*n* = 1554) were identified as being at high risk of developing type 2 diabetes. Reallocating 60 min from sitting to standing and to stepping was associated with a lower BMI, waist circumference, and triglycerides and higher high-density lipoprotein cholesterol using both ISM and CISM (*p* < 0.05). The direction and magnitude of significant associations were consistent across methods. No associations were observed for hemoglobin A1c, total cholesterol, or low-density lipoprotein cholesterol for either method. Results of both ISM and CISM were broadly similar, allowing for the interpretation of previous research, and should enable future research in order to make informed methodological, data-driven decisions.

## 1. Introduction

There is a substantial body of epidemiological evidence linking high sedentary behaviour and lack of physical activity (PA) with morbidity [1,2,3,4] and mortality [2,5,6,7,8]. Recently, there has been an increase in the number of studies that have examined both PA and sedentary behaviour and their impact or associations with health, either by modelling the association of reallocating time from one to another [9,10,11,12,13,14,15,16,17,18,19,20,21,22], or by accounting for the other behaviours as covariates [23,24,25,26].

The term ‘physical behaviours’ refers to any behaviour contributing to the 24-h movement and non-movement conceptual model outlined by Trembley et al. (2017) [27], which represents a consensus project to create clear, common, and accepted terminology and definitions. However, how to treat physical behaviour data when assessing the associations between specific behaviours and health has become a debated topic in recent years [17,28]. Often, data have been treated as continuous and unbounded, as this is the way these behaviours have typically been conceptualised. For example, when assessing the associations of time spent in sedentary behaviour with health, traditionally, one would use minutes or hours spent in sedentary behaviour as the independent variable and then include a PA variable, typically moderate-to-vigorous physical activity (MVPA), as a covariate [29]. However, by doing this, statistical models may not account for the inherent collinearity between time spent sedentary and time spent in PA. Research has shown that collinearity varies between sedentary behaviour and different intensities of PA or non-sedentary behaviours [9,29]. This has led to the development of methodological approaches which aim to address the co-dependence of physical behaviours, which are identified as composites of a finite whole (i.e., a 24-h day). Put simply, changes in one behaviour will affect time allocated to other behaviours.

Isotemporal substitution first addressed the need to recognise the finite nature of time and to understand that in order to change behaviour one must substitute time in that behaviour with time in another [30]. The isotemporal substitution model (ISM) uses data in original, conceptualised units such as hours and uses a specific time period to bind the behaviours into a finite period. Questions have been raised about the suitability of these data in absolute values (i.e., time) to be used in behavioural epidemiology [11]. It has been argued that data that are intrinsically collinear or compositional must be treated as relative values, not absolute [29]. For example, step count is an absolute value, whereas the minutes spent stepping per day are relative to the 24-h day. Compositional isotemporal substitution (CISM) seeks to address these issues for the use of absolute data, such as time spent in different activities [11,29]. It is important to understand the convergence and divergence of these two methodologies regarding associations with health to ensure previous and future evidence is correctly interpreted.

Current evidence, primarily using the ISM approach but not exclusively, has consistently shown beneficial associations with markers of health when time is reallocated from sedentary behaviour to active, non-sedentary behaviours [9,10,11,12,13,14,15,16,17,18,19,20]. Beneficial associations have been reported for post-challenge glucose and insulin [9,12,15,20], chronic-low grade inflammation [16], and other cardiometabolic biomarkers [10,13,14,15], as well as for risk of all-cause mortality [18]. A large proportion of evidence has utilised accelerometers to assess physical behaviours, typically worn at the hip or wrist [31,32], whereby sedentary time is inferred from lack of movement rather than postural allocation. A recent study noted this is a major limitation of the literature concerning sedentary behaviour and health [33]. A handful of studies utilised thigh-worn accelerometers, which have the capability of identifying posture, and modelled the reallocation of time from sitting to standing and stepping [9,12,15,16,19]. These studies all showed inverse associations with markers of health when time was reallocated from sitting to stepping, while three showed inverse associations when time was reallocated from sitting to standing [12,15,16]. Despite this, the evidence utilising thigh-worn accelerometery to examine associations between sitting and health remains limited.

This study’s aim was to use both ISM and CISM to examine and compare associations of reallocating time from sitting to standing or stepping, assessed using thigh-worn accelerometery, with markers of cardiometabolic health.

## 2. Materials and Methods

### 2.1. Participants

We performed a pooled analysis of cross-sectional data collected across 3 randomised control trials (RCTs)—the Promotion of Physical activity through structured Education with differing Levels of ongoing Support for people at high risk of type 2 diabetes (T2DM) (PROPELS), Walking Away from Diabetes, and Project STAND (Sedentary Time And Diabetes). Each study recruited individuals from primary care identified as being at high risk of developing T2DM. The protocols for each study have been published previously [34,35,36]. All studies received ethical approval and followed identical standard operating procedures (performed by different research professionals) to collect the dependent and independent variables of interest. All participants provided written informed consent. Each study had its own inclusion and exclusion criteria, as described below.

#### 2.1.1. PROPELS

The PROPELS study (ISRCTN83465245) was a multi-centre 3-arm RCT with 48-month follow-up and data collection completed in July 2019 to evaluate an intervention designed to increase physical activity in an ethnically diverse population at high risk of developing T2DM. Participants were eligible if they were aged 40–74 years and were White European, or aged 25–74 years and were South Asian, had a fasting plasma glucose (≥5.5 to <7.0 mmol/L) or HbA1c (≥6.0 to <6.5%; ≥42 to <48 mmol/mol) value within the prediabetes range [37], and had access to a mobile phone and were willing to use it as part of the study. Data collected at baseline were used for this analysis.

#### 2.1.2. Walking Away from Diabetes (WA)

The WA study (ISRCTN31392913) was a cluster RCT with 36-month follow-up completed in January 2014 which evaluated an intervention designed to increase physical activity in those at high risk of developing T2DM. Participants were eligible if they were aged 18–74 years and scored in the 90th centile of the automated version of the Leicester Risk Assessment tool [38]. Participants were excluded if they had an existing diagnosis of T2DM, were diagnosed with T2DM at a baseline appointment, were taking steroids, or were unable to speak English. Data collected at the 3-year follow-up (2013–2014) were used for this analysis as this was the only time point where activPAL data were collected (see the physical behaviour measurement section).

#### 2.1.3. STAND

The STAND study (ISRCTN08434554) was an RCT with 12-month follow-up completed in 2012 aimed at reducing sitting time in young adults at high risk of developing T2DM. Participants were eligible if they were aged 18–40 years, obese (BMI (body mass index) ≥30 kg/m^2^ for White Europeans and ≥27.5 kg/m^2^ for South Asians) or overweight (≥25 kg/m^2^ and ≥23 kg/m^2^), with 1 additional risk factor for diabetes (family history of diabetes or cardiovascular disease in a first-degree relative; previous gestational diabetes; polycystic ovarian syndrome; HbA1c ≥5.8% or ≥40 mmol/mol; impaired glucose tolerance and/or impaired fasting glucose) [39]. Data collected at baseline were used for this analysis.

### 2.2. Sedentary Behaviour and Physical Activity Measurement

Data were collected using the activPAL3™ (PAL Technologies, Glasgow, UK) attached at the midline anterior aspect of the upper thigh with a hypoallergenic dressing. The devices were waterproofed with a nitrile sleeve and wrapped in waterproof hypoallergenic dressing to allow for 24-h wear. Participants were asked to wear the device continuously for up to 7 days in PROPELS and WA and 10 days in Project STAND. The activPAL determines body posture (i.e., sitting/lying and upright activity, standing and stepping) [40]. The activPAL has been shown to be highly accurate in detecting lying, sitting, and upright behaviours [41]. activPAL data (event files) were processed using Processing PAL v1.21 (University of Leicester, UK) [42]. This java application uses a validated algorithm to identify valid waking wear time [43], and it produces summary data based on the identified valid waking wear data. The default algorithm thresholds within the application were used. The processed data were visualised using the heatmaps created within the application to identify any occasions where the algorithm incorrectly coded ‘sleep’ and waking behaviour (e.g., early wake and bedtimes in comparison to other days). On such occasions the self-reported sleep log was referred to, and if necessary, the data were corrected. Participants with ≥4 valid wear days (≥10 h, ≥500 step events (i.e., 1000 steps), ≤95% spent in sitting, standing or stepping) were included in this analysis [43].

### 2.3. Anthropometric and Blood Pressure Measurement

Blood pressure was measured in a sitting position (Omron, Healthcare, Henfield, UK). In total 3 measurements were taken, with an average of the last 2 calculated. Body weight, body fat percentage, height and waist circumference were measured to the nearest 0.1 kg, 0.5%, 0.5 cm, and 0.1 cm, respectively. Waist circumference was measured at the midline between the iliac crest and the lowest rib. Body mass index (BMI) was calculated as mass (kg)/height^2^ (m).

### 2.4. Cardiometabolic Biomarker Measurement

The cardiometabolic outcomes which were measured across all 3 studies were included in this analysis. All biomarkers were assessed by venous sampling, obtained after an overnight fast and analysed in clinical laboratories using validated quality-controlled assays. Analysis was conducted by individuals blinded to the patients’ identity using stable methodologies (the ability of the sample material to maintain its original properties) standardised to external quality assurance values. The biomarkers included were haemoglobin A1c (HbA1c) and lipid profile (total cholesterol, high-density lipoprotein (HDL) cholesterol, low-density lipoprotein (LDL) cholesterol, triglycerides). A clustered cardiometabolic risk score (CCRS) was generated to assess overall cardiometabolic risk by calculating an average of the standardised ((value–mean)/SD) values for HbA1c, triglycerides, mean of systolic and diastolic blood pressure, HDL cholesterol (inverted), and waist circumference. This method has been reported and validated previously [44,45,46]. Overall, 2 scores were calculated, with 1 including waist circumference and 1 without. This allowed us to examine any mediation effect of change in adiposity for any associations observed for the CCRS. A higher score represents a higher cardiometabolic risk.

### 2.5. Covariates

Date of birth, sex, ethnicity, and current medication were recorded using an interview-administered questionnaire. Participants who reported taking angiotensin converting enzyme inhibitors, alpha-blockers, angiotensin receptor blockers, beta-blockers, calcium channel blockers, or diuretics/thiazides were classified as taking blood pressure medication. Participants who reported taking statins or fibrates were classified as taking lipid-lowering medication. An Index of Multiple Deprivation (IMD) score was generated using the participants’ home postcodes. These scores are publicly available continuous measures of compound social and material deprivation that are calculated using a variety of data including current income, employment, education, and housing. All covariates were chosen due to their potential influence on cardiometabolic health.

### 2.6. Statistical Analysis

Analyses were performed using R statistical systems (version 3.4.3, R Foundation for Statistical Computing, Vienna, Austria) and IBM SPSS Statistics 24 (IBM, Armonk, NY, USA). ISM and CISM were conducted to examine the associations of reallocating 60 min of sitting for standing or stepping with markers of cardiometabolic health and adiposity. Dependent (outcome) variables were standardised to allow comparisons. The covariates included in both ISM and CISM were age, sex, ethnicity, IMD score, blood pressure medication, and lipid-lowering medication. Both models report β coefficients which represent a 1-unit (hour) change in a given behaviour, with significance set at an alpha at ≤0.05.

#### 2.6.1. Isotemporal Substitution Modelling

Linear regression models were conducted following an ISM [30]. For the purposes of this analysis, ISM required average waking wear time, standing time, and stepping time to be simultaneously entered into a regression model, with the resulting regression coefficient for standing time and stepping time representing the association of substituting a given unit of sitting time (in this case, 60 min) into each category, respectively. Importantly, the inclusion of average waking wear time ensures the reallocation is modelled within a given time frame—the time participants were awake.

#### 2.6.2. Compositional Isotemporal Substitution Modelling

CISM were conducted using similar methodologies as outlined previously [9,11,47]. CISM uses isometric log ratios (ILRs) to model the physical behaviour composition within real space, allowing conventional statistical models to be conducted. ILRs are calculated for various compositions modelling the reallocations of time from sitting to standing and stepping. In this case, ILRs were calculated to reflect a reallocation of 60 min from sitting to standing and for 60 min from sitting to stepping. These are calculated from the mean physical behaviour composition. These ILRs are then fitted to a multiple linear regression model, which models the reallocation of time spent in one behaviour with another pairwise within a set time frame, in this case waking time when wearing the activPAL device. There were no zero values in the combined dataset, and therefore no recoding had to be undertaken.

#### 2.6.3. Incremental Comparison

In order to assess the dose–response association for reallocating time from one behaviour to another, both ISM and CISM were run to model 5-min reallocations to 60-min reallocations at 5-min intervals. Five-minute reallocations were conducted to replicate, statistically, experimental research that has widely used 5-min changes in behaviour [48,49,50]. This resulted in 12 separate models for ISM and CISM respectively. This was conducted for zBMI only in this instance to explore the convergence and divergence between models more closely. Previous epidemiological work has shown that, when examining cardiometabolic health, the strongest and most consistent associations exist between sedentary time, physical activity, and adiposity [26].

## 3. Results

A total of 2388 participants were recruited across the PROPELS, WA, and STAND studies, of which 1524 participants had valid activPAL data and were included in these analyses (sample loss of 36%) (see Figure 1). The mean age was 59.8 years (standard deviation 11.9), 51.7% were male, 72.9% were White European, and the mean BMI was 30.3 kg/m^2^ (5.72). Basic participant characteristics are included in Table 1. Participants not included in these analyses (N = 864) were younger (58 years of age vs. 60 years of age, *p* = 0.023) and had a higher HbA1c (5.81% vs. 5.76%, *p* = 0.008).

Associations of modelling reallocation of time from sitting to standing and from sitting to stepping for both CISM and ISM are presented in Figure 2 and Figure 3. Associations are also reported in Appendix A. Overall, both models showed favourable associations when reallocating time from sitting to standing or stepping for BMI, waist circumference, triglycerides, HDL cholesterol, and CCRS (with and without adiposity). The strongest associations for sitting to standing or stepping were for waist circumference and BMI for both models. No significant associations were found for any behavioural reallocation for HbA1c, total cholesterol, or LDL cholesterol in either model.

Figure 4 shows the associations of reallocating time from sitting to standing or stepping, and from standing and stepping to sitting for zBMI. Reallocations were made in 5-min increments, starting with 5 min through to 60 min for both ISM and CISM. This demonstrates how the reallocations are asymmetrical for CIMS, yet symmetrical for ISM. Reallocating 60 min of sitting to stepping and of stepping to sitting for zBMI using ISM equates to a β-coefficient of −0.37 (−0.44, −0.29) and 0.37 (0.44, 0.29) respectively, showing perfect symmetry. Whereas, reallocating 60 min of sitting to stepping and of stepping to sitting for zBMI using CISM equates to an estimated difference of −0.23 (−0.27, −0.19) and 0.32 (0.25, 0.38) respectively, which is asymmetrical. The reallocations are also assumed to be linear for ISM, whereas for CISM they are non-linear. For example, ISM shows that the association for reallocating 30 min of sitting to stepping is exactly half the association for reallocating 60 min of sitting with stepping (−0.183 (−0.222, −0.143), −0.365 (−0.444, −0.286)). CIMS, however, shows the association for reallocating 30 min of sitting with stepping is a little more than half of the association for reallocating 60 min of sitting with stepping (−0.122 (−0.146, −0.098), −0.230 (−0.274, −0.185)). The non-linear associations for CISM are further demonstrated when reporting the reallocations from stepping to sitting. For example, the association for reallocating 30 min of stepping with sitting is 45% of the estimated difference for reallocating 60 min of stepping with sitting (0.142 (0.114, 0.170), 0.315 (0.251, 0.378)).

## 4. Discussion

This study used ISM and CISM analyses to examine the associations of replacing time spent sitting with standing or stepping and with markers of cardiometabolic health. Overall, the findings showed minimal differences in the magnitude of associations between the methods. Both methods demonstrated favourable associations for the reallocation of 60 min of sitting to standing and to stepping for BMI, waist circumference, triglycerides, HDL cholesterol, and CCRS (with and without adiposity). Nevertheless, whilst it was observed that the overall interpretation was similar between models, associations were consistently stronger for ISM models, although these differences were statistically and clinically negligible.

The main difference between methods was in the incremental comparisons, where time was reallocated from sitting to standing or stepping, and vice versa. Here we showed that CISM produced small, yet observable, asymmetrical and non-linear associations when time was reallocated at 5-min increments. ISM produced symmetrical and linear associations. Considering this, the results of modelling the reallocation of larger periods of time may result in larger differences between models, impacting the interpretation of results. However, given the negligible clinical and statistical difference in the magnitude of associations for small reallocations of time, the interpretations of these results are fundamentally similar. However, it is possible that these differences could have an impact on public health messages if the reasons for any differences between models are not better understood. It is therefore advised that future analyses should consider the potential impact of these differences in the interpretation of either ISM or CISM outputs, and perhaps use both methodologies simultaneously. Mechanistic and interventional research is needed to investigate which model provides the better estimation of the ‘true’ association. Furthermore, although minimal differences are observed between models in these analyses, it may be the case that differences are observed in other samples with different mean compositions of behaviour.

One study has compared these methodologies previously; however, this was only done for one variable (body fat percentage) and was in children [22]. Furthermore, a previous study that utilised CISM did include the results of ISM in the supplementary material [9]. However, there was no direct comparison of the results in the main paper. Previous work involving both methods have claimed the superiority of one over the other. Conversely, in the development of CISM, it was stated that ISM violates the compositional properties of time-use data [11]. Although this may be true, no evidence was provided showing the results were in some way incorrect or invalid. The results presented in this study suggest that the use of either method produces broadly comparable results when examining the reallocation of time spent sitting to standing or stepping, assessed with thigh-worn accelerometery, on markers of cardiometabolic health and adiposity. Therefore, the decision as to which methodology to use should be based on the data in use and the research question stipulated, with a consideration of the interpretation of results.

The associations observed here are similar to previous studies that used ISM and CISM to examine association of reallocating time from one physical behaviour to another on markers of cardiometabolic health [15,19,21,51]. Two previous studies that utilised thigh-worn accelerometery to assess physical behaviours showed that reallocating time from sitting to movement was favourably associated with waist circumference, triglycerides, and HDL [15,19]. Reallocating time from sitting to standing was favourably associated with triglycerides and HDL [15]. In older adults, Ryan et al. (2019) showed that modelling the reallocation of time in sedentary behaviour to time in physical activity was favourably associated with total cholesterol and triglycerides [21]. In the current study, favourable associations were observed for reallocating time from sitting to standing and stepping for triglycerides, but not for total cholesterol. Further to this, it was previously shown that, over a 7-year period, reallocating time from sedentary behaviour to MVPA was associated with a reduction in BMI and body fat percentage for older women [51]. This supports the results presented here, because the associations of the reallocation of sitting to standing and stepping were the strongest for BMI and waist circumference.

It is noteworthy that reallocation of time from sitting to standing had favourable associations. Some, but not all, experimental studies support this observation, particularly in those at risk of developing type 2 diabetes [48]. However, this study supports the majority of epidemiological studies conducted to date, which have suggested that whilst standing may have some positive associations, associations are consistently stronger for re-allocation into more intensive forms of movement [46].

This study has several strengths. The measurement of sedentary behaviour and physical activity was with a device that accurately distinguishes between postures (i.e., sitting, standing, and stepping). Widely used and valid health markers were included. The large, heterogeneous, multi-ethnic sample of individuals identified as being at high risk of T2DM accurately reflects individuals likely to receive/benefit from interventions aiming to reducing sitting by increasing movement.

However, this study is not without limitations. The sample is not necessarily generalisable to the general population, meaning results should be treated with caution in non-clinical, healthy populations. There is the potential limitation of pooling three large datasets. However, each study followed the same standard operating procedures when collecting the data. Furthermore, models were adjusted for differences between datasets (i.e., age). Results are cross-sectional; therefore, inferences about causality are not possible. However, the dataset provided a valuable resource with which to test the comparability of the two dominant statistical approaches used to model behavioural reallocation. Similar work is encouraged to replicate these findings in healthy populations across different age and sociodemographic groups.

## 5. Conclusions

In conclusion, reallocating time from sitting to standing or from sitting to stepping showed beneficial associations with BMI, waist circumference, triglycerides, HDL, and CCRS (with and without adiposity). Results from ISM and CISM are broadly similar, with no differences observed in the direction or magnitude of associations. Minor differences were observed in the symmetry of associations, which requires future research to examine the possible mechanism underpinning these differences, whether it be statistical or physiological. The differences in the symmetry of associations do suggest these methodologies may diverge to a greater extent when larger amounts of time are reallocated, therefore limiting the comparability of these methodologies to smaller reallocations of time. These results allow for appropriate and informed decisions on methodology based on a data-driven approach.

## Figures and Tables

**Figure 1 ijerph-18-06210-f001:**
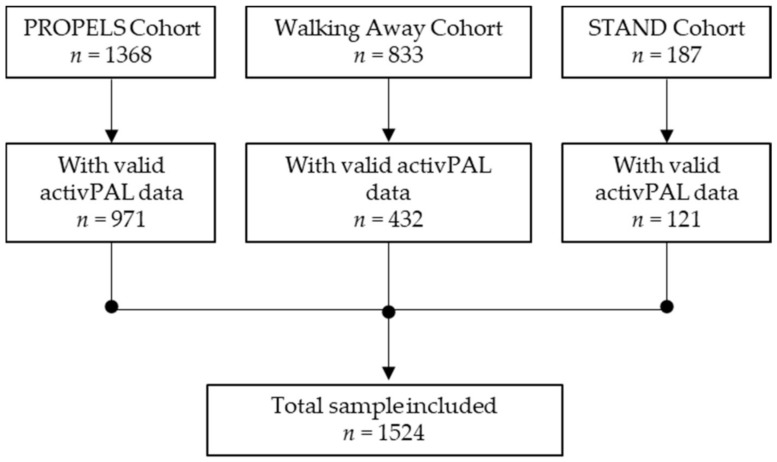
Participant flow diagram.

**Figure 2 ijerph-18-06210-f002:**
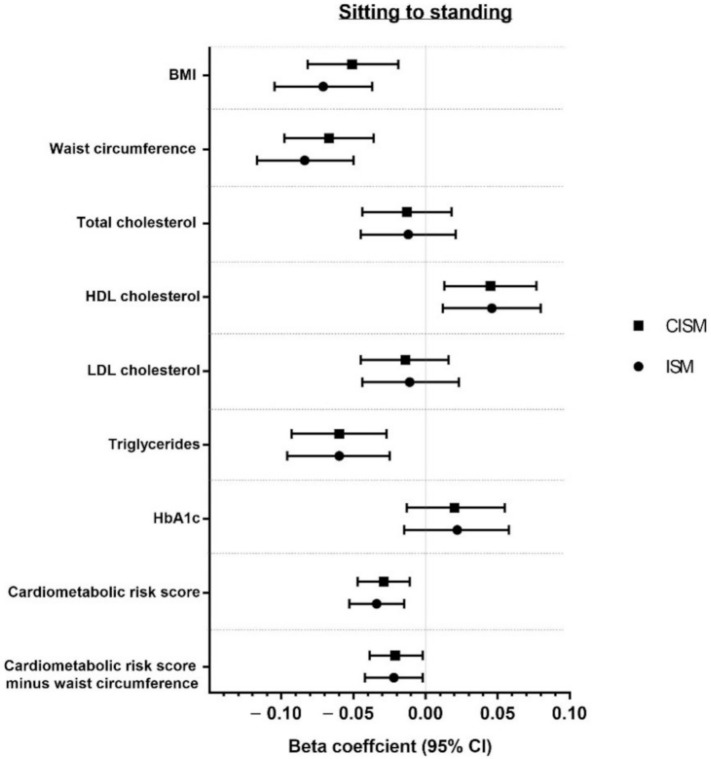
Reallocation of 60 min from sitting to standing.

**Figure 3 ijerph-18-06210-f003:**
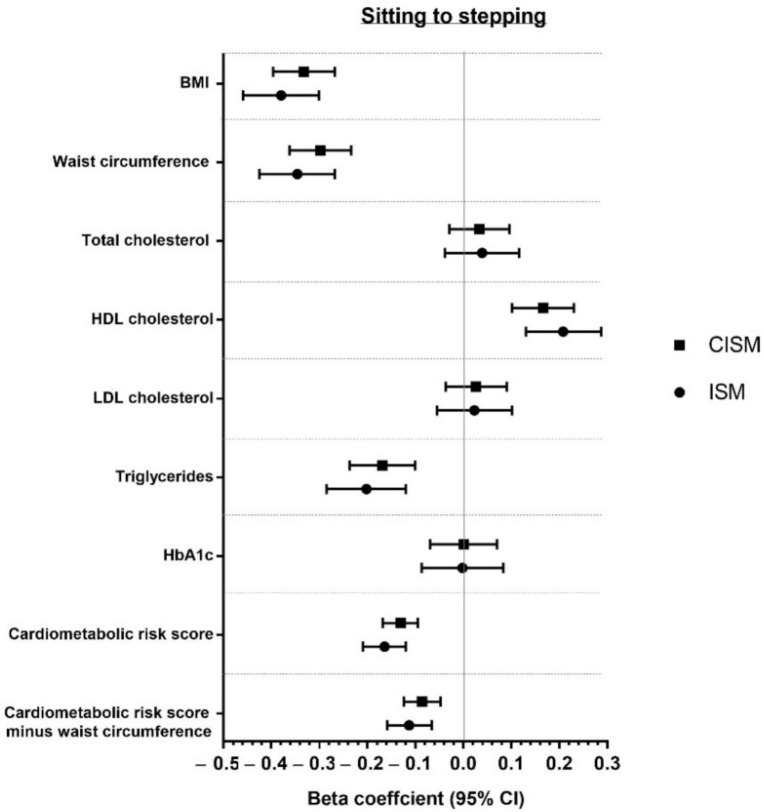
Reallocation of 60 min from sitting to stepping.

**Figure 4 ijerph-18-06210-f004:**
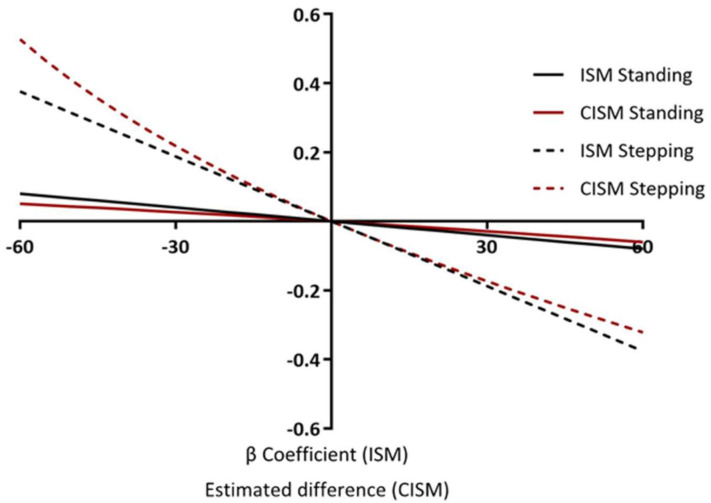
Incremental reallocations: multiple behavioural reallocations examining 5-min increments on zBMI.

**Table 1 ijerph-18-06210-t001:** Participant characteristics.

Characteristics	All (*n* = 1524)	PROPELS (*n* = 971)	WA (*n* = 432)	STAND (*n* = 121)
Age (years)	59.8 (11.9)	59.9 (9.0)	66.8 (7.4)	32.8 (5.7)
Male (%)	51.7	49.9	61.8	29.9
White European (%)	72.9	70.5	89.1	75.2
Body mass index (kg/m^2^)	30.3 (5.7)	29.3 (5.7)	31.4 (5.3)	34.4 (5.1)
Waist circumference (cm)	100 (14)	98 (14)	103 (13)	103 (13)
Using blood pressure medication (%)	40.2	38.2	48.6	3.3
Using lipid-lowering medication (%)	27.9	28.3	30.1	0.8
Current smokers (%)	9.1	8.9	7.2	19.0
HbA1c (unit %)	5.8 (0.4)	5.8 (0.3)	5.7 (0.5)	5.5 (0.3)
Sitting (min/day)	548 (112)	542 (113)	568 (109)	530 (107)
Standing (min/day)	286 (96)	296 (97)	268 (89)	277 (91)
Stepping (min/day)	107 (40)	111 (40)	101 (38)	103 (37)

HbA1c = Haemoglobin A1c, data for continuous variables are reported as mean ± standard deviation.

## Data Availability

The data presented in this study are available on request from the corresponding author. The data are not publicly available due to ethical restrictions.

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
