# Peer review of "Modelling the Reallocation of Time Spent Sitting into Physical Activity: Isotemporal Substitution vs. Compositional Isotemporal Substitution"

_ijerph, 2021, doi:10.3390/ijerph18126210_

Round 1
Reviewer 1 Report
Abstract
Lines 32 – can you change ‘informs’, to prevent informs/informed used twice in the sentence.
Introduction
Line 38 – gap after ‘haviour’.
Lines 54 – change to ‘intensities’.
Lines 56 and 58 – the ‘s’ of behaviours is separated? Not sure if this is how the draft has been sent for review??
Line 60 and 61 – gap after behaviour?
Line 68 – amend to ‘absolute data, such as time…’.
Line 74 – if ‘beneficial’ associations then reiterate this.
Materials and Methods
Lines 99-101 – mention different researchers performed the testing to be transparent, then support with as you have stated …it followed identical procedures etc. Maybe around here state different time points for measures were used and different acceptance criteria. Then that leads it the three separate studies.
Line 104 – ‘data collection was completed by…’.
Line 113 – amend to read something similar to ‘which evaluated an intervention designed…’.
Line 161 – can you expand on ‘stable methodologies’ and ‘external quality assurance values’.
Line 197 – could flow better, ‘…, when the participants are awake’ for example.
Line 206 – amend first ‘models’ to ‘model’.
Line 211 – ‘from’ not ‘form’.
In section 2.6.3 you mention modelling 5-minute reallocations to 60-minute reallocations. Can you support this with literature as to why you have taken this approach? For example, why 5-minutes and not 10 minutes?
Results
Figures 2 and 3 are reallocation of 30 minutes from sitting to standing and stepping, respectively, not 60 minutes? Throughout the method (sections 2.6.1 and 2.6.2) you have been mentioning 60 minutes then suddenly 30 minutes has been presented. I think this needs to be clearer in the method section or figures showing 60 minutes of reallocation in the results section. Or, in the results a limited amount of text stating why 30 minutes is shown in the figures.
Line 246 – the first ‘minutes’ change to ‘minute’.
Discussion
Line 284-285 – I think you could make your point clearer when you mention ‘if not better understood’.
Line 289 – ‘Furthermore’ needs spelling correctly.
Line 313/314 and 320 – ‘favourable’.
Line 318 – comma after ‘here’.
Line 323 – ‘suggested’.
Reviewer 2 Report
Dear authors,
Thanks for the well written manuscript.
I was very happy with the overall presentation and content, and liked the comparison made between the methods, as there has been discussion on them.
I only noted a technical error throughout the document that split words in few places or created unnecessary large spaces between words. I also saw misspelled words in some other way, Examples:
Page 2, lines 56 and 58 behaviour s...
Page 8, line 289: ...Furthermoire...
Page 9, line 339 ...behaviour al...
Please check the entire document once more for such errors.
Reviewer 3 Report
Introduction
Generally, the introduction is weak to address the study purpose such as ‘to use both ISM and CISM to examine and compare associations of reallocating time from sitting to standing … cardiometabolic health’.
Line 40-41, Please rephrase this sentence to improve understanding or, if possible, remove or relocate the sentence. It is my sole opinion that line 42 should be the leading sentence.
Line 44, please add more citations in addition to 27 to jusify that the definition is very common.
Line 51, please add appropriate citations.
Line 52-53. I think that this sentence is very important. However, there is no supplementary or strong information avaiable to justify the statement because I may be stating that by doing this statistical model may account for the colinearity between time spent sedentary and time spent in PA.
Line 48-58, the statement from 52-53 contradicts the statement from 57-58.
Method
Please state differences and similarities of the participants in three different studies.
Line 99 need IRB number.
Line 171-180 It would be important to identify covariates for this study. Please state reasons or logics for choosing covariates.
Line 190-197 Please provide an example of computation for the both models to allow readers to see differences or similarities of the two models.
Reviewer 4 Report
Dear Authors,
It was a pleasure to read your work. You can find amendments in enclosed file.
Kind regards,
Reviewer

Round 2
Reviewer 3 Report
At this time, I do not have any comment.
Author Response
Thank you for your time reviewing our manuscript.